# AD-Reasoning: Multimodal Guideline-Guided Reasoning for Alzheimer's Disease Diagnosis

## Abstract

Accurate diagnosis of Alzheimer's disease (AD) requires combining multimodal data with established clinical guidelines. However, most deep learning models operate as black boxes, offering limited interpretability and weak alignment with medical standards. We propose AD-Reasoning, a framework for multimodal AD diagnosis that integrates structural MRIs and diverse clinical data with guideline-guided reasoning. A rule engine ensures NIA-AA diagnostic criteria, while reinforcement fine-tuning with domain-informed rewards promotes clinically consistent and transparent decision-making. Evaluated on the AD-MultiSense dataset, AD-Reasoning achieves state-of-the-art diagnostic accuracy and demonstrates improved interpretability compared with recent baselines. This work highlights a clinically grounded solution that connects large language models with medical expertise, advancing interpretable and guideline-compliant AD diagnosis.

## 1 Introduction

The rapid advancement of artificial intelligence (AI) has profoundly impacted neurodegenerative disease research, showing great promise in medical data analysis and diagnostic applications Rajpurkar et al. (2022); Park et al. (2023). In the context of Alzheimer's disease (AD), many existing studies focus on single-modal data, most commonly structural magnetic resonance imaging (sMRI) Frisoni et al. (2010); Jang & Hwang (2022) or individual clinical assessments Öhman et al. (2021). Although such approaches can be effective within specific domains, they often offer a narrow view that overlooks AD's complex and multifactorial pathology. In reality, AD spans a wide range of physiological and behavioral manifestations: brain atrophy patterns visible in sMRI, cognitive decline quantified by neuropsychological tests (e.g., MMSE), genetic risk factors such as APOE-$\epsilon$4, cerebrospinal fluid (CSF) biomarkers (e.g., Abeta42, pTau), as well as demographic information, comorbidities, and lab findings Lautner et al. (2014). This heterogeneity underscores the limitations of single-modality models, which may yield incomplete or biased diagnostic conclusions. To address this, comprehensive multimodal integration is essential for a more holistic understanding and accurate characterization of AD Venugopalan et al. (2021).

Recent advances have explored multimodal fusion for AD diagnosis, integrating information from neuroimaging, clinical assessments, genetic markers, and biochemical indicators Chen et al. (2024); Zhou et al. (2023). Although these approaches enhance diagnostic performance, they typically function as black-box models, yielding only binary labels or scalar scores without offering transparent reasoning or detailed justification. These shortcomings become particularly critical in complex clinical scenarios, e.g., differentiating AD from overlapping neurodegenerative conditions. The absence of interpretable and text-based diagnostic rationales hinders clinical adoption, as physicians require not only accurate decisions but also an understanding of the underlying evidence to inform treatment and build trust in AI-assisted tools.

Multimodal large language models (MLLMs) OpenAI (2023); Grattafiori et al. (2024) have recently emerged as a powerful paradigm, demonstrating strong capabilities in cross-modal representation alignment and generative reasoning. In the medical domain, early efforts have applied MLLMs to unimodal tasks, such as automated sMRI reporting Bai et al. (2024), clinical text summarization, or single-modality image captioning. However, these models are typically constrained to surface-level physiological descriptions within individual modalities, falling short of producing coherent diagnostic narratives grounded in multimodal clinical evidence. Crucially, they lack the ability to

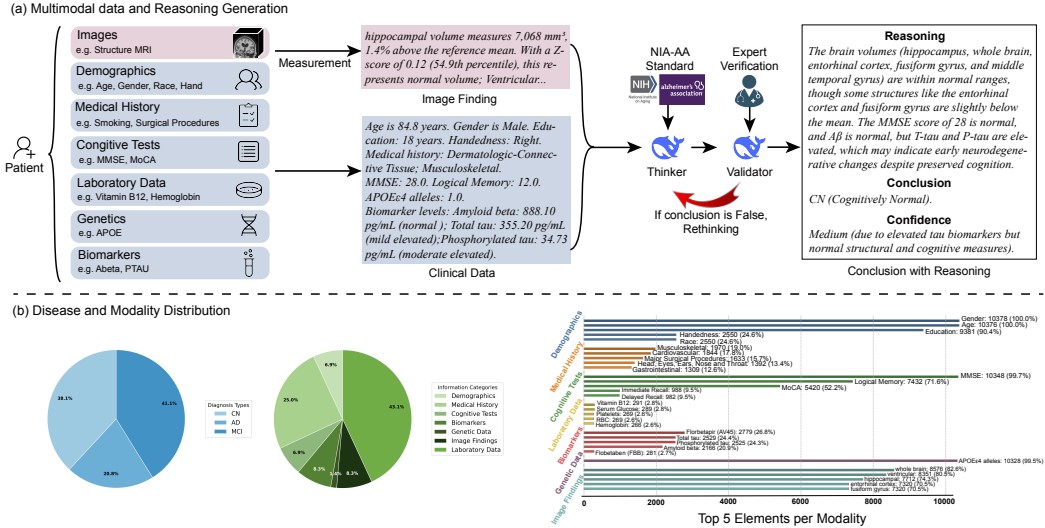

Figure 1: Our AD-MultiSense dataset. (a) Construction pipeline: Disease-level reports are generated via evidence-augmented reasoning, using DeepSeek-v3 under clinical guidelines with self-refinement for diagnostic validity. (b) Data statistics: The dataset covers CN, MCI, and AD cases, spanning seven modalities (demographics/history/cognition/labs/genetics/biomarkers/sMRI).

perform disease-level reasoning, e.g., distinguishing overlapping pathologies or integrating diverse risk factors, based on synergistic understanding across imaging, clinical, and molecular data. A unified MLLM framework that can synthesize heterogeneous patient data into interpretable, multi-disease diagnostic narratives remains an open and critical challenge.

To bridge this gap, we introduce AD-Reasoning, a novel MLLM framework tailored for interpretable reasoning and diagnosis of Alzheimer's disease. Given a patient's sMRI and six categories of clinical data, including demographics, medical history, cognitive assessments, laboratory tests, genetic risk factors, and CSF biomarkers, AD-Reasoning generates clinically grounded diagnostic narratives that integrate heterogeneous evidence. To tackle the challenge of aligning heterogeneous inputs from imaging and diverse clinical sources, we design a modality-aware encoder that projects all modalities into a shared latent space while preserving semantic fidelity. We further introduce a multimodal fusion layer that explicitly models cross-modal interactions and adaptively estimates the contribution of each modality. This design enables the model to focus on salient clinical cues, facilitating more accurate differential diagnosis and comorbidity reasoning. In addition, diagnostic narratives should be not only accurate but also consistent with clinical guidelines and expert logic. To this end, we introduce a domain-specific reinforcement learning (RL) stage, leveraging Group Relative Policy Optimization (GRPO) and a clinical consistency reward that encourages the model to generate trustworthy and guideline-aligned explanations.

Our main contributions are as follows:

- **AD-MultiSense Dataset**: We build the first AD-specific multimodal question-answer (QA) dataset combining sMRI with six clinical modalities, totaling 10,378 entries from 2,619 subjects. QA pairs span both physiological understanding and diagnostic reasoning, validated via NIA-AA criteria and expert-in-the-loop sampling.

- **AD-Reasoning Framework**: We propose a unified multimodal reasoning model that features a modality-harmonized encoder, a cross-modal fusion and reasoning layer for comorbidity-aware inference, and a domain-aligned reinforcement fine-tuning scheme that enhances interpretability and clinical consistency.

- **State-of-the-art Performance**: Our *AD-Reasoning* achieves strong results on AD diagnosis, comorbidity differentiation, and interpretable report generation, validated across large-scale multisite cohorts.

## 2 RELATED WORKS

**MLLM for Medical Diagnosis** The diagnostic potential of MLLMs stems from their proficiency in handling varied inputs, such as text Haltaufderheide & Ranisch (2024), images Chen et al. (2023), tabular data Fang et al. (2024). Early approaches were modality-specific, focusing on clinical text Van Veen et al. (2024), medical imaging Tian et al. (2023), or single biomarkers Elsborg & Salvatore (2023). Despite this progress, AD research remains siloed, with sMRI analysis largely separated from critical clinical information like cognitive tests, genetics, and biomarkers Yao et al. (2023). While emerging multimodal frameworks tackle general diagnostic fusion Kumar et al. (2024), none are designed for AD's distinct challenge: the essential integration of sMRI findings with multifaceted clinical data to achieve comorbidity-sensitive diagnosis. In contrast, our AD-Reasoning introduces a unified MLLM that performs cross-modal interaction and contribution-aware fusion, enabling structured and stage-aware reasoning aligned with clinical criteria.

**RL for Medical Diagnosis** Group Relative Policy Optimization (GRPO) Shao et al. (2024) enhances reinforcement fine-tuning by normalizing rewards across response groups, demonstrating superiority over PPO Schulman et al. (2017) in text Hu (2025) and vision-language tasks Li et al. (2025). Recent medical applications deploy GRPO for unimodal objectives like radiology reporting Dai et al. (2025). Its utility for intricate multimodal Alzheimer's Disease (AD) diagnosis, however, remains unexamined, particularly regarding: (1) reward design: Existing functions (e.g., Jaccard similarity) fail to capture clinical validity in AD diagnostics. (2) multimodal grounding: Limited work integrates GRPO with multimodal data fusion. (3) reasoning verification: Absence of NIA-AA-aligned reward mechanisms for diagnostic chains. We pioneer GRPO adaptation for AD via a clinical consistency reward function, explicitly optimized for 1) adherence to NIA-AA diagnostic criteria, 2) accuracy in comorbidity reasoning and, 3) faithfulness to multimodal evidence chains. This ensures generated diagnostic reports are both statistically robust and clinically verifiable.

## 3 METHODOLOGY

### 3.1 AD-MULTISENSE DATASET

**Multimodal Data Collection** To enable MLLMs to perform both physiological understanding and diagnostic reasoning over heterogeneous medical data, we construct a multimodal dataset that conforms to established clinical logic. Raw data are collected from the ADNI Petersen et al. (2010) and AIBL Ellis et al. (2009) cohorts, covering a wide spectrum of patient characteristics and disease stages. For each subject, we acquire sMRI scans alongside six types of clinical data encompassing demographic, cognitive, and biochemical information. After aligning data across modalities and visit timepoints, we curate a total of 10,378 multimodal samples from 2,619 unique subjects. Each sample reflects a consistent physiological state at a specific visit, enabling clinically valid reasoning over disease progression.

To enhance clinical interpretability, quantitative measurements are systematically converted into standardized textual reports. For sMRI analysis, we calculate age-adjusted $z$-scores for structural volumes (e.g., hippocampal/ventricular) using population norms, with textual descriptors generated based on established thresholds: bilateral hippocampus atrophy is reported as "mild" ($1 \leq |z| < 1.5$), "moderate" ($1.5 \leq |z| < 2$), "significant" ($2 \leq |z| < 3$) or "profound" ($|z| \geq 3$). Similarly, laboratory data undergoes $z$-score normalization against age/sex-matched cohorts, though only clinically significant abnormalities ($|z| > 2.0$) are included in final reports. Biomarkers are consistently interpreted with contextual information, and each value is accompanied by reference-based interpretation, e.g., "Amyloid beta: 858.30 pg/mL (normal)." This quantitative-to-textual transformation bridges raw biomarker measurements with clinically meaningful narratives, enabling natural language reasoning about pathological changes while preserving data fidelity. Dataset statistics are visualized in Fig. 1(b), and implementation details, including z-score normalization and templated text construction, are provided in Appendix A.

**Reasoning Generation** Based on these raw data, we construct multimodal QA pairs from disease-level diagnostic reasoning, with the entire process shown in Fig. 1(a). The process begins by querying the *Thinker* model (DeepSeek-V3) using a structured diagnostic prompt template:

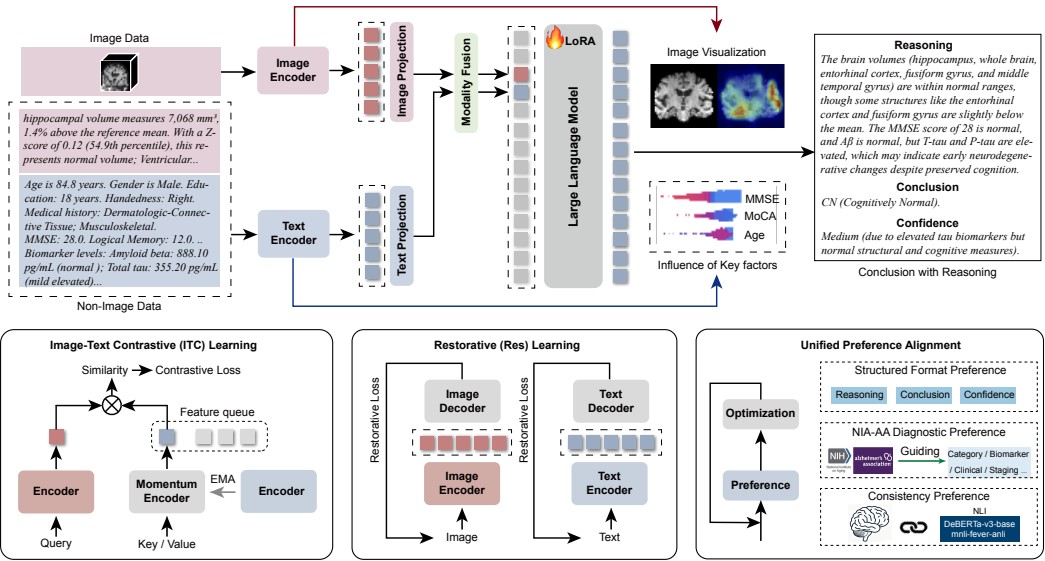

Figure 2: AD-Reasoning framework. Pretraining aligns sMRI and clinical data representations via encoders, SFT tunes LLMs using diagnostic rationales and RFT optimizes with GRPO for NIA-AA compliant structured outputs.

```
SYSTEM_PROMPT: "You are an Alzheimer's specialist.  Analyze
the data and provide:
1.  Reasoning
2.  Final diagnosis:  CN/MCI/Dementia
3.  Confidence level:  High/Medium/Low
Format:
Reasoning:  [analysis]
Diagnosis:  [CN/MCI/Dementia]
Confidence:  [High/Medium/Low]"
```

This is an initial response $\langle R_0, C_0 \rangle = \text{Thinker}(M, P_d)$, where $R_0$ denotes the reasoning chain, $C_0$ is the preliminary diagnosis, $M$ represents multimodal inputs (i.e., sMRIs and clinical data), and $P_d$ is the diagnosis prompt.

The *Validator* module evaluates $C_0$ against ground truth diagnoses. When mismatches occur, the system triggers rethinking cycles: the Thinker regenerates reasoning using refinement prompts ($P_r$) constructed from explicit NIA-AA criteria dictionaries. These dictionaries map clinical findings to diagnostic rules, enabling targeted feedback. This iterative process continues for up to N cycles (i.e., 2), with random expert sampling providing quality control.

For cases where diagnosis remains incorrect after N iterations, the prompts with correct diagnosis ($P_c$) is explicitly provided to the Thinker, instructing it to correct its reasoning and conclusion accordingly. The Thinker then produces final reasoning $R^F$ and diagnosis $C^F$, formatted into training pairs $\langle M \circ P_d, R^F \circ C^F \rangle$ for supervised fine-tuning.

### 3.2 AD-REASONING FRAMEWORK

#### 3.2.1 MODEL ARCHITECTURE

The proposed AD-Reasoning framework primarily consists of modality-specific encoders and projectors, a Multimodal Fusion Layer (MFL), and a Large Language Model (LLM), with its overall architecture illustrated in Fig. 2. Given the raw data of structural MRI scans $\mathbf{X_V} \in \mathbb{R}^{1 \times D \times H \times W}$ and clinical text data $\mathbf{X_T} \in \mathbb{R}^L$, they are first processed by their respective modality-specific encoders for feature extraction. The encoded features are then fed into modality-specific projectors to

transform them into a shared dimension $d$ for alignment and compatibility with the textual embedding space of the LLM. This process facilitates seamless integration between multimodal features and textual tokens, formulated as:

$$\mathbf{V}_{\text{sMRI}} = g_V(f_V(\mathbf{X_V})) \in \mathbb{R}^d, \quad \mathbf{T}_{\text{Clinical}} = g_T(f_T(\mathbf{X_T})) \in \mathbb{R}^d, \tag{1}$$

where $\mathbf{V}_{\text{sMRI}}$ denotes projected visual features from structural MRI, $\mathbf{T}_{\text{Clinical}}$ denotes projected clinical text features. $f_V, f_T$ denotes modality-specific encoders (image and text) and $g_V, g_T$ denotes modality-specific projectors.

### 3.2.2 MULTIMODAL FUSION LAYER (MFL)

To enable comprehensive interaction between neuroimaging and clinical modalities, we introduce an MFL comprising a Bidirectional Cross-Attention (BCA) mechanism. The projected features $\mathbf{V}_{\text{sMRI}}$ and $\mathbf{T}_{\text{Clinical}}$ are first processed by the BCA mechanism, where each modality alternately serves as Query and Key/Value to compute cross-attention:

$$\mathbf{A}_{V \rightarrow T} = \text{Attention}(\mathbf{T}_{\text{Clinical}}, \mathbf{V}_{\text{sMRI}}, \mathbf{V}_{\text{sMRI}}), \tag{2}$$

$$\mathbf{A}_{T \rightarrow V} = \text{Attention}(\mathbf{V}_{\text{sMRI}}, \mathbf{T}_{\text{Clinical}}, \mathbf{T}_{\text{Clinical}}). \tag{3}$$

This bidirectional attention captures complex neuro-clinical dependencies, allowing visual features to inform clinical interpretation and vice versa. The attention outputs are combined with residual connections to preserve modality-specific information:

$$\mathbf{T}_V = \mathbf{V}_{\text{sMRI}} + \mathbf{A}_{T \rightarrow V}, \quad \mathbf{T}_T = \mathbf{T}_{\text{Clinical}} + \mathbf{A}_{V \rightarrow T}. \tag{4}$$

### 3.2.3 LARGE LANGUAGE MODEL INTEGRATION

The final multimodal features $\mathbf{T}_V$ and $\mathbf{T}_T$ replace the placeholders `<sMRI>` and `<clinical>` in the input prompt templates. An example prompt for AD diagnosis is:

> "Given the structural MRI `<sMRI>` and clinical profile `<clinical>`, what is the most probable diagnosis and supporting evidence?"

The resulting input sequence $\mathbf{T}_{\text{input}} = \{\mathbf{T_Q}, \mathbf{T}_V, \mathbf{T}_T, \mathbf{T_A}\}$ is fed into the LLM, where $\mathbf{T_Q}$ denotes tokenized question derived from diagnostic templates and $\mathbf{T_A}$ denotes target answer tokens from AD diagnostic QA datasets.

The LLM parameters remain frozen during training, with only LoRA adapters updated to specialize the model for AD reasoning tasks.

### 3.2.4 TRAINING STRATEGY

We employ a three-stage training strategy for AD-Reasoning, which includes Pre-training (PT), Supervised Fine-Tuning (SFT), and Reinforcement Fine-Tuning (RFT), to progressively enhance its ability to perceive the physiological representations of each modality and integrate multimodal information for interpretable alzheimer's disease reasoning and diagnosis.

**Pre-training (PT).** To establish foundational understanding and align feature representations across imaging and non-imaging clinical data, we first conduct pre-training using AD-relevant multimodal data. During this stage, the image encoder (processing sMRI) and text encoder (processing clinical data) are trainable, while projectors and LLM parameters remain inactive at this stage. The optimization focuses exclusively on representation learning and alignment.

We employ the image-text contrastive (ITC) loss Radford et al. (2021) to align image features $h_I$ and text features $h_T$ generated by the image and text encoders. The ITC loss $\mathcal{L}_{\text{itc}}$ maximizes similarity for positive image-text pairs while suppressing negative pairs, implemented through normalized cross-entropy over all pairwise similarities. We implement momentum encoders updated via exponential moving average (EMA) following BLIP Li et al. (2022) and ALBEF Li et al. (2021). Specifically, the parameters of momentum image/text encoders ($\xi$) are updated as $\xi \leftarrow m_c \cdot \xi + (1 - m_c) \cdot \theta$, where $m_c = 0.995$ is the momentum coefficient and $\theta$ denotes the parameters of the corresponding online encoders. All momentum encoders operate without gradient backpropagation. This EMA-based strategy ensures feature consistency within the dynamically updated data and knowledge

| | Method | BLEU | METEOR | ROUGE | BERT | ACC (%) | AUC (%) | SEN (%) | SPE (%) |
|---|---|---|---|---|---|---|---|---|---|
| CN vs. CI | LLaVA-1.5-7B | 0.0112 | 0.1456 | 0.1023 | 0.7924 | 73.85 | 68.92 | 60.14 | 80.37 |
| | LLaVA-Med | 0.0144 | 0.1618 | 0.1168 | 0.8016 | 76.21 | 71.43 | 62.75 | 83.42 |
| | Med-PaLM-M | 0.0218 | 0.2031 | 0.1331 | 0.8181 | 79.92 | 75.76 | 66.63 | 85.85 |
| | M3d-LaMed | 0.0341 | 0.1756 | 0.1435 | 0.8128 | 82.37 | 78.95 | 69.84 | 86.21 |
| | AD-Reasoning w/o PT | 0.1873 | 0.2792 | 0.2424 | 0.8636 | 87.25 | 83.12 | 71.28 | 91.37 |
| | AD-Reasoning w/o RFT | 0.2015 | 0.2982 | 0.2617 | 0.8725 | 90.46 | 87.63 | 80.75 | 94.28 |
| | AD-Reasoning (ours) | **0.2183** | **0.3212** | **0.2851** | **0.8926** | **93.33** | **91.83** | **88.67** | **95.00** |
| CN vs. MCI | LLaVA-1.5-7B | 0.0108 | 0.1387 | 0.0984 | 0.7821 | 70.15 | 65.28 | 61.42 | 74.85 |
| | LLaVA-Med | 0.0138 | 0.1518 | 0.1068 | 0.7916 | 72.24 | 68.76 | 65.57 | 77.36 |
| | Med-PaLM-M | 0.0208 | 0.1931 | 0.1231 | 0.8081 | 75.13 | 72.14 | 68.41 | 80.25 |
| | M3d-LaMed | 0.0331 | 0.1656 | 0.1335 | 0.8028 | 78.02 | 74.97 | 70.79 | 81.64 |
| | AD-Reasoning w/o PT | 0.1824 | 0.2717 | 0.2369 | 0.8570 | 88.37 | 84.96 | 84.92 | 87.41 |
| | AD-Reasoning w/o RFT | 0.1961 | 0.2893 | 0.2544 | 0.8667 | 91.28 | 89.07 | 88.45 | 90.33 |
| | AD-Reasoning (ours) | **0.2123** | **0.3125** | **0.2783** | **0.8852** | **92.82** | **90.09** | **88.60** | **93.50** |

Table 1: Comparison of AD-Reasoning and baselines in terms of reasoning and diagnostic performance for Alzheimer's disease.

queues by decoupling momentum encoder optimization from the online model training. To prevent abrupt shifts in feature distribution, the queues are exclusively maintained using outputs from the momentum encoder.

Our restorative learning module is designed to enhance the global semantic understanding by incorporating fine-grained visual and textual information. That is, the feature extraction is augmented by a reconstruction learning branch, which includes an image decoder to reconstruct the original image from the representation and minimizes the pixel-level distance between the original image $x_I$ and the reconstructed image $x'_I$: $\mathcal{L}^I_{res} = \mathbb{E}_{x_I} \mathcal{D}_I(x_I, x'_I)$, where $\mathcal{D}_I(x_I, x'_I)$ presents the distance function that measures similarity between $x_I$ and $x'_I$, e.g., Mean Square Error (MSE), or L1 norm. We use MSE following the common setting He et al. (2022). For the textual component, we apply a similar approach. A text decoder is trained to minimize the token-level distance between the original text $x_T$ and the reconstructed text $x'_T$: $\mathcal{L}^T_{res} = \mathbb{E}_{x_T} \mathcal{D}_T(x_T, x'_T)$, where $\mathcal{D}_T(x_T, x'_T)$ is the distance function measuring text similarity, such as the commonly-used cross-entropy loss.

The overall pre-training objective combines both alignment and reconstruction losses:

$$\mathcal{L}_{\text{PT}} = \mathcal{L}_{\text{itc}} + \lambda_{\text{res}} \left( \mathcal{L}^I_{res} + \mathcal{L}^T_{res} \right) \tag{5}$$

where $\mathcal{L}_{\text{itc}}$ denotes image-text contrastive loss for feature alignment, $\mathcal{L}^I_{res}$ denotes image reconstruction loss (MSE), $\mathcal{L}^T_{res}$ denotes text reconstruction loss (cross-entropy), $\lambda_{\text{res}}$ denotes weighting coefficient for reconstruction objectives.

**Supervised Fine-Tuning (SFT).** Building upon the aligned feature representations, we conduct SFT using diagnostic QA pairs for AD reasoning. During this stage, image and text encoders are frozen, and the projection layers and LLM LoRA modules are trainable. The optimization objective maximizes response generation likelihood:

$$\mathcal{L}_{\text{SFT}} = -\mathbb{E}_{(\mathbf{T_Q}, \mathbf{V}_{\text{sMRI}}, \mathbf{T}_{\text{Clinical}}, \mathbf{T_A}) \sim \mathcal{D}} \cdot \sum_{t=1}^{T} \log \pi_\theta \left( y_t \mid \mathbf{T_Q}, \mathbf{V}_{\text{sMRI}}, \mathbf{T}_{\text{Clinical}}, y_{<t} \right), \tag{6}$$

where $\pi_\theta(y_t|\cdot)$ denotes the conditional probability of generating the $t$-th token $y_t$, given the prompt tokens $\mathbf{T_Q}$, modality features ($\mathbf{V}_{\text{sMRI}}$ and $\mathbf{T}_{\text{Clinical}}$), and the previously generated tokens $y_{<t}$. $\mathbf{V}_{\text{sMRI}}$ denotes visual features from structural MRI and $\mathbf{T}_{\text{Clinical}}$ encompasses all clinical texts. $\mathbf{T_Q}$ denotes question tokens and $\mathbf{T_A}$ denotes answer tokens.

**Reinforcement Fine-Tuning (RFT).** To unlock the potential of the constructed dataset and enhance diagnostic reasoning capabilities, we perform Reinforcement Fine-Tuning (RFT) using Group Relative Policy Optimization (GRPO) under the RL with Verifiable Rewards (RLVR) framework. The trainable components remain consistent with the SFT stage, with the optimization objective:

$$\max_{\pi_\theta} \mathbb{E}_{\mathbf{A} \sim \pi_\theta(\mathbf{Q})} \left[ R_{\text{RLVR}}(\mathbf{Q}, \mathbf{A}) \right] = \left[ R(\mathbf{Q}, \mathbf{A}) - \beta \, \text{KL} \left[ \pi_\theta(\mathbf{A} \mid \mathbf{Q}) \| \pi_{\text{ref}}(\mathbf{A} \mid \mathbf{Q}) \right] \right] \tag{7}$$

where $\pi_\theta$ is the policy and $\pi_{\text{ref}}$ is the SFT-tuned reference. $R$ denotes the verifiable reward function, while the KL divergence term penalizes deviation from clinically validated responses, with $\beta$ controlling the regularization strength.

| Method | CN vs. CI | | | | CN vs. MCI | | | |
|---|---|---|---|---|---|---|---|---|
| | ACC | AUC | SEN | SPE | ACC | AUC | SEN | SPE |
| BERT | 84.31 | 79.42 | 85.87 | 86.48 | 82.55 | 77.35 | 82.67 | 84.42 |
| RoBerta | 86.89 | 84.41 | 82.97 | 85.03 | 85.63 | 81.42 | 80.93 | 83.84 |
| Longformer | 87.92 | 85.76 | 80.49 | 82.27 | 85.24 | 84.71 | 78.42 | 79.37 |
| IRENE | 86.03 | 77.95 | 89.14 | 65.82 | 84.18 | 75.25 | 87.35 | 63.27 |
| AD-Trans | 87.67 | 75.89 | 65.91 | 85.47 | 85.61 | 73.79 | 63.67 | 84.32 |
| Alifuse | 87.23 | 79.51 | **90.71** | 73.67 | 85.98 | 76.57 | **88.92** | 70.39 |
| Ours | **93.33** | **91.83** | 88.67 | **95.00** | **92.82** | **90.09** | 88.60 | **93.50** |

Table 2: Diagnostic performance (%) comparison between our AD-Reasoning and classification approaches for Alzheimer's disease. (Best in bold)

| Task | ACC | AUC | SEN | SPE |
|---|---|---|---|---|
| **(a) Loss Terms** | | | | |
| $\mathcal{L}_{\text{itc}}$ | 89.23 | 84.87 | 95.12 | 79.84 |
| $\mathcal{L}_{\text{itc}} + \mathcal{L}_{\text{res}}^{I} + \mathcal{L}_{\text{res}}^{T}$ | **93.33** | **91.83** | **88.67** | **95.00** |
| **(b) Feature Terms** | | | | |
| Image | 71.24 | 54.76 | 95.33 | 12.31 |
| Clinical | 88.83 | 82.69 | 96.91 | 67.42 |
| Image + Clinical | **93.33** | **91.83** | **88.67** | **95.00** |
| **(c) Guideline Terms** | | | | |
| IWG-2 | 92.93 | 90.58 | **90.12** | 87.33 |
| NIA-AA | **93.33** | **91.83** | 88.67 | **95.00** |

Table 3: Ablation results (%) on the test set.

For AD diagnosis where responses exhibit high clinical specificity, GRPO directly compares responses within candidate groups $\{o_1, \ldots, o_G\}$. Reward normalization uses: $\tilde{r}_i = \frac{r_i - \mu_r}{\sigma_r + \epsilon}$, where $\mu_r$ and $\sigma_r$ are group reward statistics. This prioritizes clinically coherent responses without requiring separate critic models.

The composite reward $R = R_F + R_{\text{NIA-AA}} + R_{\text{consistency}}$ ensures diagnostic accuracy and structural consistency:

*1) Structured Format Reward ($R_F$):* Enforces compliance with AD diagnostic templates:

```
Reasoning: [analysis]
Diagnosis: [CN/MCI/Dementia]
Confidence: [High/Medium/Low]
```

$R_F = 1.0$ only when all three tags are present and `Confidence` contains valid value.

*2) NIA-AA Diagnostic Reward ($R_{NIA-AA}$):* Provides comprehensive clinical assessment through a multi-dimensional scoring framework that evaluates diagnostic accuracy against established NIA-AA standards. The reward integrates three core components:

$$R_{\text{NIA-AA}} = 0.4 \cdot R_{\text{cat}} + 0.3 \cdot R_{\text{bio}} + 0.3 \cdot R_{\text{feat}}. \tag{8}$$

**Diagnostic Category Alignment ($R_{\text{cat}}$)** ensures precise classification into standardized diagnostic categories (CN, MCI, Dementia) through keyword matching and exclusion criteria validation. This component evaluates both the presence of appropriate diagnostic terminology and the absence of contradictory indicators.

**Biomarker Consistency Assessment ($R_{\text{bio}}$)** quantifies the coverage and contextual accuracy of essential AD biomarkers ($A\beta$, tTau, pTau). The scoring incorporates both mention frequency and pathological status characterization (normal/abnormal patterns) based on established clinical thresholds.

**Clinical Feature Comprehensiveness ($R_{\text{feat}}$)** evaluates the depth of cognitive domain analysis across memory, executive function, visuospatial abilities, and language domains. The scoring rewards not only feature inclusion but also detailed characterization within specific subdomains.

This structured approach ensures rigorous adherence to NIA-AA diagnostic protocols while maintaining computational efficiency through weighted component integration. Detailed scoring algorithms are provided in Appendix B.

*3) Reasoning Consistency Reward ($R_{consistency}$):* To ensure logical coherence between diagnostic reasoning and final conclusions, we introduce a reasoning consistency reward in the overall reward structure. This component evaluates the alignment between the analysis in the Reasoning section and the diagnostic conclusion:

$$R_{\text{consistency}} = \text{NLI}(\text{Reasoning} \rightarrow \text{Diagnosis}) \tag{9}$$

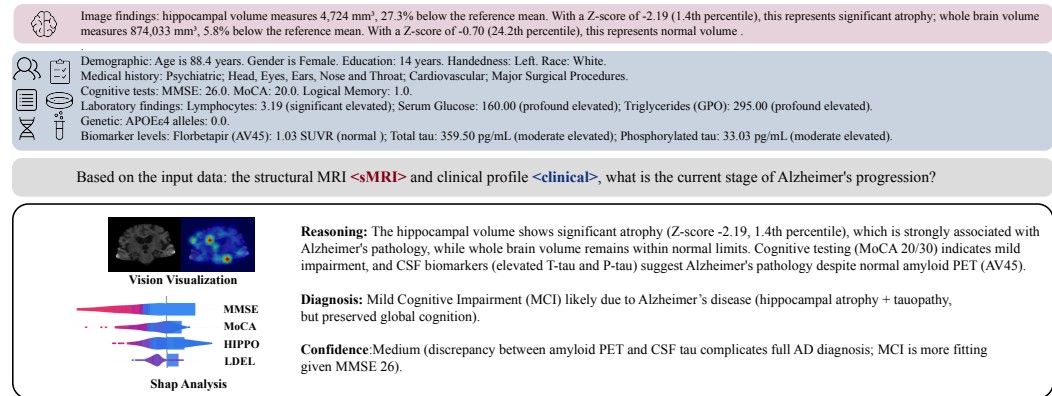

Figure 3: Inference example of AD-reasoning.

where NLI denotes Natural Language Inference He et al. (2021), implemented using a pre-trained entailment model that scores the degree to which the reasoning text supports the diagnostic conclusion. The reward $R_{\text{consistency}} \in \{0, 0.5, 1.0\}$ corresponds to contradiction, neutral/weak entailment, and strong entailment, respectively. This prevents logical inconsistencies where, for example, the reasoning describes normal biomarker profiles but concludes with "Dementia," ensuring that diagnostic conclusions are well-supported by the preceding clinical analysis.

This enhanced reward structure ensures comprehensive alignment with NIA-AA diagnostic standards while maintaining computational efficiency and logical coherence. The format reward $R_F$ guarantees structural integrity, $R_{\text{NIA-AA}}$ evaluates clinical content validity, and $R_{\text{consistency}}$ ensures logical alignment between analysis and conclusions.

## 4 EXPERIMENTS

We conduct all experiments on a server equipped with four NVIDIA RTX 3090 24GB GPUs. For the LLM, we choose LLaMA 3.2-1B et al. (2024) and integrate the LoRA modules Hu et al. (2022) with a rank of 8 for fine-tuning. For the visual modality, a 3D Vision Transformer Dosovitskiy et al. (2020) is used with input size $128 \times 128 \times 128$ and patch size $16 \times 16 \times 16$. For the textual modality, we use a Longformer Transformer Beltagy et al. (2020). The PT, SFT and RFT stages are each trained for 100 epochs, while the RFT stage is trained using the open-source Trainer framework.

The effectiveness of multi-disease reasoning and diagnosis is evaluated from two sides. 1) The descriptive accuracy of the generated diagnostic text is assessed using natural language generation (NLG) metrics, including BLEU, METEOR, ROUGE, and BERT. 2) The classification accuracy of Alzheimer disease categories in the responses is evaluated using diagnosis accuracy (ACC), Area Under Curve (AUC), sensitivity (SEN), and specificity (SPE).

Following established clinical guidelines McKhann et al. (2011); Dubois et al. (2007); Jack Jr et al. (2018), we evaluate our model on two classification tasks. The first task distinguishes cognitively normal (NC) individuals from those with cognitive impairment (CI), including both mild cognitive impairment (MCI) and Alzheimer's disease (AD). The second one focuses on differentiating NC from MCI, which is a critical stage for the early identification of AD. We split the dataset *subject-wise* into training, validation, and test sets with proportions of 70%, 10%, and 20%, respectively. All structural MRI scans underwent standardized preprocessing, including skull stripping Isensee et al. (2019) to remove non-brain tissues and intensity normalization to harmonize voxel value distributions across scanners.

### 4.1 QUANTITATIVE ANALYSIS

Given the absence of multimodal models specialized for AD integrating neuroimaging and comprehensive clinical data, we adapt comparative frameworks by measuring sMRI volumes and generating

descriptions, representing clinical profiles as structured text narratives. Table 1 benchmarks AD-Reasoning against four state-of-the-art MLLMs: LLaVA-1.5-7B Liu et al. (2023), LLaVA-Med Li et al. (2023), Med-PaLM-M Tu et al. (2024) and M3D-LaMed Bai et al. (2024). These models represent the current frontier in medical multimodal reasoning.

As shown in Table 2, to evaluate the performance of our model, we select three prominent text-only baselines(e.g., BERT Devlin et al. (2018), Roberta Liu (2019), and Longformer Beltagy et al. (2020)) and three recent transformer-based models that fuse multimodal information for classification(e.g., IRENE Zhou et al. (2023), AD-Trans Yu et al. (2024), and Alifuse Chen et al. (2024)).

The results demonstrate that AD-Reasoning outperforms these leading models, excelling not only in natural language generation but also in clinical evaluation. This indicates the superior capability of AD-Reasoning in both descriptive and diagnostic reasoning tasks in multi-disease scenarios. Furthermore, Table 1 also presents ablation studies to investigate the impact of physiological-level pre-training and RFT-based post-training on the model's performance. The results show that removing either component leads to a noticeable decline in performance. Specifically, the findings highlight two key insights: 1) Pre-training enables the model to extract and align high-quality, modality-specific representations while preserving fine-grained information through restoration loss, establishing a robust foundation for cross-modal reasoning. 2) The RFT stage based on GRPO further unleashes the potential of the constructed data and enhances the model's multi-disease diagnostic performance, enabling deeper and more effective cross-modal reasoning.

### 4.2 QUALITATIVE ANALYSIS AND ABLATION STUDY

AD-Reasoning demonstrates a robust ability to integrate and analyze data from multiple modalities to arrive at comprehensive diagnoses. This integration allows for mutual corroboration among the modalities, enhancing diagnostic accuracy, as shown in Fig. 3. AD-Reasoning effectively synthesizes information from sMRI and clinical non-image data to diagnose alzheimer's disease conditions. Each modality provides unique insights that collectively strengthen the diagnostic conclusion. The model frequently employs terms, e.g.,"indicates" and "associated with", highlighting its capability to identify and utilize evidence from each modality to substantiate the final diagnosis. This approach demonstrates AD-Reasoning's proficiency in extracting relevant features from each dataset, ensuring that the diagnostic reasoning is well-founded and comprehensive. To enhance interpretability, we apply Shapley analysis Lundberg & Lee (2017) on test sets to identify the most influential numerical features in diagnostic decisions, and implement the method from Chefer et al. (2021) to visualize attention heatmaps in the visual encoder. More details can be found in Appendix C.

The ablation studies in Table 3 demonstrate the effectiveness of both contrastive and restorative learning modules in the pre-training, as well as the necessity of complete modality integration. The integration of $\mathcal{L}_{\text{itc}}$ and $\mathcal{L}_{\text{res}}$ significantly enhances the results, validating our initial intention to design these mechanisms to facilitate modality fusion and adjust the contribution levels of different modalities for various diseases. The presence of all modalities results in the best performance. Removing any single modality leads to reduced scores. This underscores the importance of multimodal integration for optimal outcomes.

## 5 CONCLUSION

In this paper, we propose a novel framework, AD-Reasoning, which represents a significant advancement in multimodal reasoning for Alzheimer's Disease diagnosis. By integrating structural MRI with comprehensive clinical data (demographics, medical history, cognitive tests, lab results, genetics, and biomarkers), AD-Reasoning overcomes the limitations of unimodal approaches and enables holistic neuro-clinical assessment. The novel AD-MultiSense dataset facilitates precise diagnostic reasoning through quantitative-to-textual transformation and NIA-AA guided refinement. The clinical-guided fusion mechanism ensures context-aware interpretation of neuroimaging findings, while Reinforcement Fine-Tuning with Group Relative Policy Optimization and NIA-AA verifiable rewards enhances diagnostic precision and reliability. Extensive validation demonstrates AD-Reasoning's superior performance in both neurophysiological understanding and differential diagnosis, highlighting its potential for real-world clinical applications, including early detection and progression monitoring in cognitive disorders.

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

## A  VISION DESCRIPTION GENERATION

The vision description generation module transforms quantitative neuroimaging measurements into clinically interpretable natural language descriptions. This transformation employs a multi-step analytical process that contextualizes individual volumetric data within population-based reference distributions. For each brain structure of interest, the system first establishes an age and gender-matched reference cohort derived from cognitively normal subjects. This cohort is stratified into decade-wide age groups (50-59, 60-69, 70-79, 80-89 years) with separate distributions maintained for male and female populations.

Three core metrics are computed to quantify deviations from normative values. The Z-score represents standard deviation units from the reference mean, calculated as

$$Z = (V_{subject} - \mu_{ref})/\sigma_{ref} \tag{10}$$

where $V_{subject}$ is the observed volume, $\mu_{ref}$ is the reference mean, and $\sigma_{ref}$ is the reference standard deviation. The percentile rank indicates the proportion of healthy individuals with smaller volumes, derived from the cumulative distribution function of the reference population. The percentage difference expresses relative deviation as

$$\Delta\% = (V_{subject} - \mu_{ref})/\mu_{ref} \times 100, \tag{11}$$

providing an intuitive measure of volumetric change.

Clinical severity classifications incorporate structure-specific pathological directionality. For atrophy-sensitive structures including the hippocampus, entorhinal cortex, fusiform gyrus, middle temporal gyrus, and whole brain, we apply the criteria in Table 4:

Table 4: Clinical interpretation of Z-scores for brain structures

| Z-score Range | Clinical Interpretation |
|---|---|
| $Z < -3$ | Profound atrophy |
| $-3 \leq Z < -2$ | Significant atrophy |
| $-2 \leq Z < -1.5$ | Moderate atrophy |
| $-1.5 \leq Z < -1$ | Mild atrophy |
| $-1 \leq Z \leq 1$ | Normal volume |
| $1 < Z \leq 1.5$ | Mild enlargement |
| $1.5 < Z \leq 2$ | Moderate enlargement |
| $2 < Z \leq 3$ | Significant enlargement |
| $Z > 3$ | Profound enlargement |

These thresholds align with established radiological practice while maintaining statistical rigor.

Natural language generation follows a standardized template that synthesizes these quantitative metrics into clinically actionable interpretations for all six structures. Each description includes four key elements: 1) the absolute volumetric measurement, 2) percentage difference from the reference mean, 3) Z-score with corresponding percentile rank, and 4) clinical severity assessment. The template dynamically adapts terminology based on pathological directionality, using "below" and "atrophy" for cortical structures versus "above" and "enlargement" for ventricles. This approach ensures consistent reporting while maintaining clinical relevance across diverse brain structures.

Table 5: Representative vision descriptions for brain structures

| Structure | Generated Description |
|---|---|
| Ventricles | Ventricular volume measures 42,500 $mm^3$, 32.5% above the reference mean (32,070 ± 2,850 $mm^3$). With a Z-score of 3.65 (99.9th percentile), this represents significant enlargement. |
| Hippocampus | Hippocampal volume measures 2,850 $mm^3$, 28.2% below the reference mean (3,970 ± 350 $mm^3$) for this demographic. The Z-score of -3.21 (0.1th percentile) indicates significant atrophy. |
| Whole Brain | Whole brain volume measures 950,000 $mm^3$, 8.7% below the reference mean (1,040,000 ± 45,000 $mm^3$). The Z-score of -2.00 (2.3th percentile) demonstrates mild atrophy. |
| Entorhinal Cortex | Entorhinal cortex volume is 2,350 $mm^3$, 35.1% below reference values. The Z-score of -3.02 (0.1th percentile) is consistent with significant atrophy. |
| Fusiform Gyrus | Fusiform gyrus volume measures 18,600 $mm^3$, 15.3% below the reference mean (21,970 ± 1,850 $mm^3$). With a Z-score of -1.82 (3.4th percentile), this suggests mild atrophy. |
| Middle Temporal Gyrus | Middle temporal gyrus volume measures 17,600 $mm^3$, 22.7% below the reference mean (22,750 ± 2,100 $mm^3$). The Z-score of -2.45 (0.7th percentile) demonstrates significant atrophy. |

Table 5 presents representative outputs of the vision description generation system for all six brain structures. These structured interpretations provide clinicians with immediately actionable information by contextualizing quantitative measurements within population norms. The comprehensive coverage of ventricles, hippocampal formation, global brain volume, and temporal lobe structures enables a holistic assessment of neurodegenerative patterns. The framework's modular design permits seamless integration of additional brain regions while maintaining standardized reporting protocols across neuroimaging evaluations.

Figure 5 presents a comparative analysis of six key brain structure volumes across diagnostic groups: cognitively normal (CN), mild cognitive impairment (MCI), and Alzheimer's disease dementia (AD/Dementia). Violin and box plots demonstrate significant volumetric differences in all structures that effectively discriminate between diagnostic categories. Most notably, ventricular volume

Figure 4: Distribution of hippocampal Z-scores across demographic and clinical dimensions.

exhibits progressive enlargement across the CN→MCI→AD continuum, while hippocampal, entorhinal, and mid-temporal volumes show corresponding stepwise reductions. Fusiform and whole brain volumes similarly decrease with disease progression. The distributions reveal three critical patterns: 1) AD patients consistently demonstrate the most pronounced atrophy (or ventricular expansion), 2) MCI subjects exhibit intermediate values with greater distributional overlap with both CN and AD groups, and 3) CN individuals maintain the highest preserved volumes. These z-score distributions provide robust imaging biomarkers that collectively differentiate diagnostic categories, with ventricular and hippocampal measures showing the most distinct group separation.

Figure 4 presents a comprehensive analysis of hippocampal volume Z-scores, normalized to age- and gender-matched cognitively normal references. Panel A shows the overall distribution with clinically significant thresholds at Z = -1 and Z = -2, revealing a right-skewed distribution indicative of hippocampal atrophy in the cohort. The boxplot analysis in Panel B demonstrates progressive Z-score reduction across the diagnostic continuum (CN → MCI → Dementia), with females exhibiting consistently lower Z-scores than males within each diagnostic category ($\Delta Z$ = [gender-diff], p $\leq$ 0.001).

Panel C illustrates the interaction between aging and neurodegeneration, where dementia patients show substantially lower Z-scores across all age groups, particularly in the 70-79 cohort. The scatterplot in Panel D confirms the expected age-related decline in hippocampal volumes (r = [correlation-value], p $\leq$ [p-value]), while highlighting the diagnostic separation maintained across the age spectrum. The horizontal reference lines at Z = -1 and Z = -2 provide clinical context for interpreting individual data points.

## B  NIA-AA Diagnostic Reward Function Specification

The NIA-AA diagnostic reward function provides a comprehensive assessment framework for evaluating Alzheimer's disease diagnostic reports generated by our model. This multi-dimensional scoring system ensures clinical accuracy and adherence to established NIA-AA diagnostic standards through three core components with weighted integration:

$$R_{\text{NIA-AA}} = 0.4 \cdot R_{\text{category}} + 0.3 \cdot R_{\text{biomarker}} + 0.3 \cdot R_{\text{feature}} \tag{12}$$

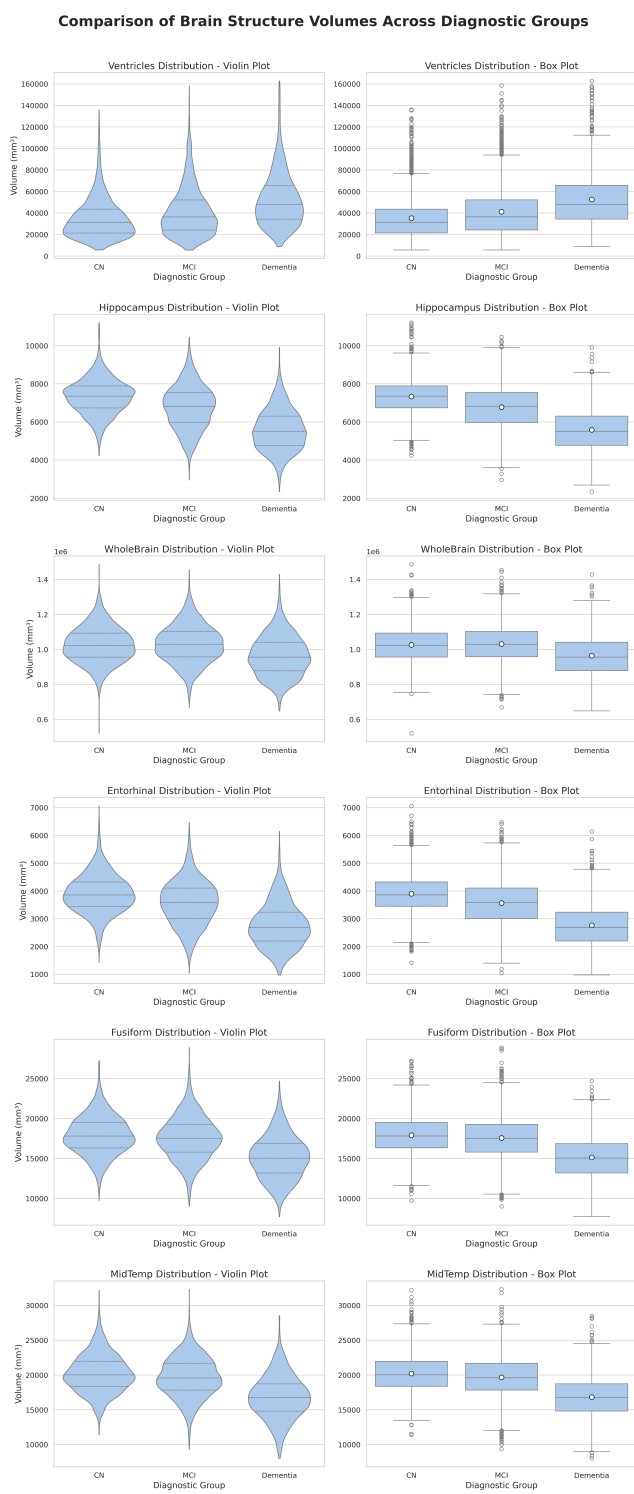

Figure 5: Volumetric distributions of six brain structures across diagnostic groups. Left column: Violin plots showing density distributions and quartiles. Right column: Box plots with white circles indicating means. Structures shown (top to bottom): Ventricles, Hippocampus, WholeBrain, Entorhinal, Fusiform, and MidTemp. CN = Cognitively Normal (n=2732), MCI = Mild Cognitive Impairment (n=3150), Dementia = Alzheimer's Disease Dementia (n=1349). Note progressive ventricular enlargement and hippocampal/entorhinal atrophy across the CN→MCI→AD continuum.

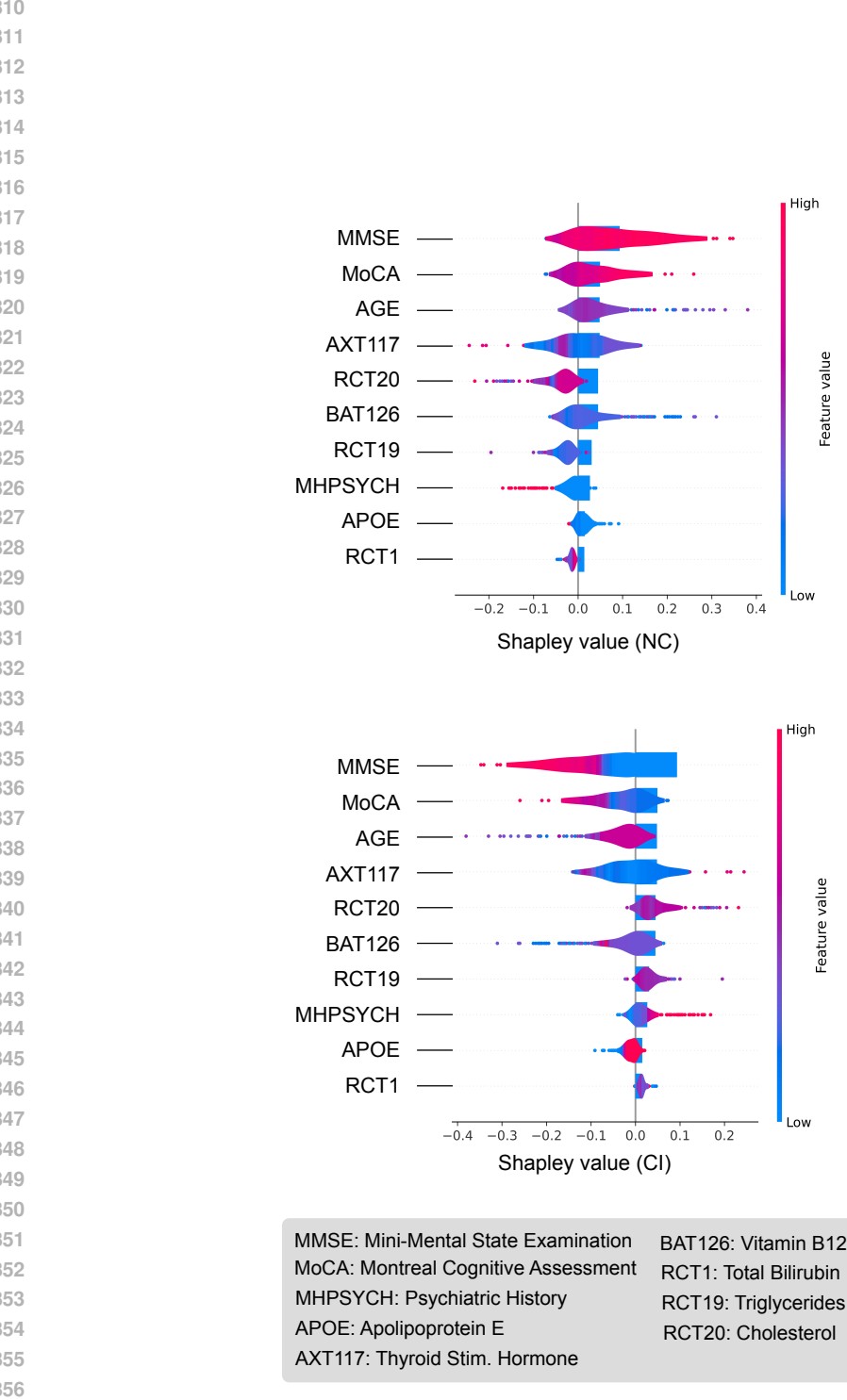

Figure 6: Shapley analysis.

## B.1  DIAGNOSTIC CATEGORY MATCHING ($R_{\text{CATEGORY}}$)

The diagnostic category component evaluates the accuracy of diagnostic classification through multi-tiered keyword validation. This 40%-weighted component ensures precise alignment with standard diagnostic categories (CN, MCI, Dementia) while penalizing contradictory terminology.

The scoring incorporates inclusion validation and exclusion penalty mechanisms:

$$R_{\text{category}} = \mathbb{I}\text{inclusion} \cdot (1 - \mathbb{I}\text{exclusion}) + R_{\text{staging}} \tag{13}$$

where $\mathbb{I}$inclusion validates presence of category-appropriate keywords, $\mathbb{I}$exclusion penalizes contradictory terminology, and $R_{\text{staging}}$ provides additional scoring for dementia stage assessment.

## B.2  BIOMARKER CONSISTENCY ($R_{\text{BIOMARKER}}$)

The biomarker consistency component (30% weight) evaluates both coverage and pathological characterization of core AD biomarkers (A$\beta$, pTau, tTau). The assessment employs clinical importance weighting and status consistency validation.

The scoring formula integrates mention frequency and status accuracy:

$$R_{\text{biomarker}} = \sum_{b \in \mathcal{B}} w_b \cdot (\alpha \cdot \mathbb{I}\text{mention}(b) + \beta \cdot \mathbb{I}\text{status}(b)) \tag{14}$$

where $\mathcal{B} = \text{A}\beta, \text{pTau}, \text{tTau}$ represents the biomarker set, $w_b$ denotes clinical weights ($w_{\text{A}\beta} = 0.4$, $w_{\text{pTau}} = 0.3$, $w_{\text{tTau}} = 0.3$), $\mathbb{I}$mention detects biomarker presence, and $\mathbb{I}$status evaluates pathological status consistency.

Status assessment utilizes pattern recognition for normal/abnormal classification:

$$\mathbb{I}\text{status}(b) = \frac{\sum p \in P_b^{\text{normal}} \mathbb{I}(p) + \sum_{p \in P_b^{\text{abnormal}}} \mathbb{I}(p)}{|P_b^{\text{normal}} \cup P_b^{\text{abnormal}}|} \tag{15}$$

where $P_b$ represents status-indicative patterns for biomarker $b$.

## B.3  CLINICAL FEATURE COVERAGE ($R_{\text{FEATURE}}$)

Clinical feature assessment (30% weight) evaluates cognitive domain coverage across memory, executive function, visuospatial abilities, and language domains. The scoring incorporates both breadth of coverage and descriptive specificity with clinical significance weighting.

The comprehensive scoring framework:

$$R_{\text{feature}} = \sum_{f \in \mathcal{F}} w_f \cdot (\gamma \cdot \mathbb{I}\text{domain}(f) + \delta \cdot \mathbb{I}\text{specificity}(f)) \tag{16}$$

where $\mathcal{F} = \text{memory, executive, visuospatial, language}$ represents cognitive domains, $w_f$ denotes clinical significance weights, $\mathbb{I}$domain evaluates primary domain coverage, and $\mathbb{I}$specificity assesses subdomain characterization depth.

Domain-specific weighting reflects clinical importance in AD diagnosis:

$$w_f = \begin{cases} 0.4 & \text{memory} \\ 0.3 & \text{executive function} \\ 0.2 & \text{visuospatial abilities} \\ 0.1 & \text{language} \end{cases} \tag{17}$$

### B.4 TEXT PROCESSING PIPELINE

The reward function employs a robust text processing workflow including format sanitization, case normalization, and clinical tokenization. Structured field extraction utilizes regular expression patterns:

$$\text{Diagnosis} = \text{extract}(\text{response}, \langle\text{diagnosis}\rangle.*?\langle/\text{diagnosis}\rangle) \tag{18}$$

$$\text{Reasoning} = \text{extract}(\text{response}, \langle\text{reasoning}\rangle.*?\langle/\text{reasoning}\rangle) \tag{19}$$

This algorithmic framework ensures rigorous adherence to NIA-AA diagnostic protocols while maintaining computational efficiency through weighted component integration. The implementation provides clinically meaningful reward signals that guide the reinforcement learning process toward generating accurate, comprehensive, and logically consistent AD diagnostic reports.

## C SHAPLEY ANALYSIS

Shapley analysis Lundberg & Lee (2017) is performed on the test sets to identify the clinical numerical features that most significantly influenced the model's diagnostic decisions (Fig. 6). The MMSE score consistently ranks among the most influential features. Thyroid Stimulating Hormone, Vitamin B12 levels, and the presence of APOE4 alleles are selected consistently among the top ten factors. These findings align with clinical studies that emphasize the strong association of MMSE scores and other key biomarkers with cognitive impairment and AD diagnosis.

