# OpenReview forum: "AD-Reasoning: Multimodal Guideline-Guided Reasoning for Alzheimer’s Disease Diagnosis"
_ICLR.cc/2026/Conference — ICLR 2026 Conference Withdrawn Submission_

### Official Review · Reviewer_zwwL · 2025-10-16

**Soundness:** 2
**Presentation:** 3
**Contribution:** 2
**Rating:** 4
**Confidence:** 3

**Summary:**

The paper presents an interesting idea and is generally well written; however, the clinical validity and reliability of the proposed dataset remain a major concern.

**Strengths:**

Clear writing and Structured Presentation.

An effort to create the novel multimodal dataset.

**Weaknesses:**

[Ground truth Leakage]
"For cases where diagnosis remains incorrect after N iterations, the prompts with correct diagnosis (Pᶜ) is explicitly provided to the Thinker, instructing it to correct its reasoning and conclusion accordingly. The Thinker then produces final reasoning R_F and diagnosis C_F, formatted into training pairs ⟨M ∘ P_d, R_F ∘ C_F⟩ for supervised fine-tuning.”";“After aligning data across modalities and visit timepoints, we curate a total of 10,378 multimodal samples… enabling clinically valid reasoning over disease progression.”
Here is the main  process of this paper from what I understand:

1.C₀ = the model’s initial diagnosis prediction.
2.The “Validator” checks C₀ against the ground truth diagnosis from ADNI/AIBL.
3.If the model is wrong after two rethinking cycles → they inject the correct label (Pᶜ) directly into the prompt.
4.The Thinker then re-generates reasoning text (R_F) and an updated diagnosis (C_F).
5.This revised reasoning + diagnosis pair is added to the training dataset used for supervised fine-tuning (SFT).
These steps may lead to the fact that "every reasoning sample in the dataset is post-hoc rationalization of the label, not independent inference."
Please rectify me if I understand wrong.

**Questions:**

Please rectify my understanding so that I could change my mind about the existing posthoc issue.

---

### Official Review · Reviewer_2PhP · 2025-11-02

**Soundness:** 3
**Presentation:** 3
**Contribution:** 3
**Rating:** 4
**Confidence:** 4

**Summary:**

The paper introduces AD-Reasoning, a multimodal large language model (MLLM) framework for interpretable Alzheimer’s Disease diagnosis. It integrates structural MRI data with six types of clinical information. A new dataset, AD-MultiSense, is developed from ADNI and AIBL cohorts, containing 10,378 samples from 2,619 subjects with multimodal QA pairs. The framework employs modality-specific encoders, a cross-modal fusion layer, and a reinforcement fine-tuning stage using Group Relative Policy Optimization (GRPO) with clinically informed rewards based on NIA-AA guidelines.

**Strengths:**

1. Innovative multimodal integration: Combines imaging, genetics, biomarkers, and clinical data into a single interpretable reasoning framework.
2. Clinical grounding: Incorporates NIA-AA diagnostic criteria through reinforcement fine-tuning, enhancing clinical validity and explainability.
3. Dataset contribution: The AD-MultiSense dataset is a valuable resource for future multimodal reasoning research in AD.
4. Strong experimental design: comprehensive baselines, quantitative metrics and ablation studies substantiate the method’s robustness.

**Weaknesses:**

1. Limited generalizability: Evaluation focuses mainly on ADNI and AIBL datasets; external validation on unseen cohorts or clinical trials is missing.
2. Potential bias in data representation: The AD-MultiSense dataset may reflect demographic imbalance including overrepresentation of certain racial groups.
3. Interpretability limits: Although textual rationales and heatmaps are presented, human expert evaluation of interpretability quality is not quantitatively assessed.

**Questions:**

How does AD-Reasoning perform on non-AD dementias? Could it generalize to multi-disease reasoning?

---

### Official Review · Reviewer_ThUN · 2025-11-02

**Soundness:** 3
**Presentation:** 4
**Contribution:** 2
**Rating:** 4
**Confidence:** 3

**Summary:**

The paper proposes AD-Reasoning, an Alzheimer Diagnosis framework that aims to integrate both MRI imaging data with other patient clinical modalities such as demographics information, lab data… into an LLM-based reasoning framework that utilizes GRPO to make the reasoning process adheres to professional diagnostic guidelines through a rule-based reward system.
In detail, the clinical modalities and MRI imaging are encoded by separate encoders, before being projected to a same semantic space and fused with cross-modal attention, and used as input to an LLM engine (DeepSeek-V3). This LLM engine (dubbed Thinker) generate reasoning-based diagnosis, and a separate LLM (Validator) provides feedback to the generated content based on various professionally defined standards for the Thinker to refine their answer in a N-step cycles. This model achieves state of the art diagnostic and reasoning results compared with other baselines, while maintaining interpretability across different modalities and decision process. In addition, the paper also contributed AD-Multisense dataset, for AD-specific Multimodal Question Answering.

**Strengths:**

1) Well-designed framework
The overall methodology is technically sound. The verbalization of text measurements to natural language is logical and intuitive, and the pretraining tasks designed to encourage multimodal alignment are proven to work effectively (Image-Text Contrastive Loss and Restorative Learning). The core part of the paper, which is the Reinforcement Learning with GRPO framework, helps evaluate the LLM's reasoning process to be factual, logical, and adhere to established clinical standards. Overall, the model's design is very efficient and logical.

2) Well-written content
The paper is generally well-written. The authors clearly outline its contributions compared with existing research, and these claims are supported with analysis and experimental results later on. The main methodology section is written logically and easy to follow, supported by eye-catching and easy to understand figures.

3) Significant experimental results, both in terms of accuracy and interpretability. The model's design also effectively allow better interpretability through Shapley Analysis, which is demonstrated in the experimental section.

**Weaknesses:**

1) Lack of novelty
While AD-Reasoning is technically impressive and achieve great accuracy, the proposed method doesn't have clear originality. It applies established techniques to the particular problem of Alzheimer Detection. The paper implements the popular and established GRPO framework, which has been applied to the medical domain [1,2] (although not Alzheimer Diagnosis in particular). The multimodal mechanism is well-designed, yet no modules stand out in terms of originality, from verbalizing quantitative measures to natural language description [3], cross-attention multimodal fusion [4,5] or popular pre-trained objectives for multimodal data like image-text contrastive loss [6,7] or restorative learning [8]. The paper needs to articulate what are the research gaps of the existing methods for each component choice: GRPO framework, the idea of textual verbalization [3], cross-attention fusion [4,5], or restorative learning [8]. And how the proposed component differs from the existing works, and how they are superior.

2) Better Assessment of LLM Reasoning
The paper reports metrics such as BLEU, METEOR or BERTScore to evaluate the quality of LLM generated reasoning. Yet, these metrics are often not very reflective of what clinicians consider "good" or "bad", and mostly seems to cover how similar in natural language the reported content is compared with the ground truth (which is in no way similar to actually factuality and logic). Direct evaluation from clinicians is required to make any conclusions in this category.


[1] Dai, W., Chen, P., Ekbote, C. and Liang, P.P., 2025. QoQ-Med: Building Multimodal Clinical Foundation Models with Domain-Aware GRPO Training. arXiv preprint arXiv:2506.00711.
[2] Zhu, W., Dong, X., Li, X., Qiu, P., Chen, X., Razi, A., Sotiras, A., Su, Y. and Wang, Y., 2025. Toward Effective Reinforcement Learning Fine-Tuning for Medical VQA in Vision-Language Models. arXiv preprint arXiv:2505.13973.
[3] Tuan Dung Nguyen, Thanh Trung Huynh, Minh Hieu Phan, Quoc Viet Hung Nguyen, and Phi Le Nguyen. 2024. CARER - ClinicAl Reasoning-Enhanced Representation for Temporal Health Risk Prediction. In Proceedings of the 2024 Conference on Empirical Methods in Natural Language Processing, pages 10392–10407, Miami, Florida, USA. Association for Computational Linguistics.
[4] Lu, J., Batra, D., Parikh, D. and Lee, S., 2019. Vilbert: Pretraining task-agnostic visiolinguistic representations for vision-and-language tasks. Advances in neural information processing systems, 32.
[5] Li, J., Li, D., Savarese, S. and Hoi, S., 2023, July. Blip-2: Bootstrapping language-image pre-training with frozen image encoders and large language models. In International conference on machine learning (pp. 19730-19742). PMLR.
[6] Li, Junnan, Dongxu Li, Caiming Xiong, and Steven Hoi. "Blip: Bootstrapping language-image pre-training for unified vision-language understanding and generation." In International conference on machine learning, pp. 12888-12900. PMLR, 2022.
[7] Li, J., Selvaraju, R., Gotmare, A., Joty, S., Xiong, C. and Hoi, S.C.H., 2021. Align before fuse: Vision and language representation learning with momentum distillation. Advances in neural information processing systems, 34, pp.9694-9705.
[8] He, K., Chen, X., Xie, S., Li, Y., Dollár, P. and Girshick, R., 2022. Masked autoencoders are scalable vision learners. In Proceedings of the IEEE/CVF conference on computer vision and pattern recognition (pp. 16000-16009).

**Questions:**

What is the training and inference cost of the framework?
What is the method's applicability to other clinical tasks and other domains which require strict and factual reasoning?

---

### Note · Authors · 2025-11-17

I have read and agree with the venue's withdrawal policy on behalf of myself and my co-authors.